# *Haor*-Based Floodplain-Rich Freshwater Ichthyofauna in Sylhet Division, Bangladesh: Species Availability, Diversity, and Conservation Perspectives

Fanindra Chandra Sarker [1,†], Md. Khaled Rahman [2,*,†], Md. Ashfaq Sadat [3], Al Shahriar [1] and A. K. M. Nowsad Alam [1]

1   Department of Fisheries Technology, Bangladesh Agricultural University, Mymensingh 2202, Bangladesh
2   Bangladesh Fisheries Research Institute, Riverine Sub-Station, Rangamati 4500, Bangladesh
3   Department of Fisheries Management, Bangladesh Agricultural University, Mymensingh 2202, Bangladesh
*   Correspondence: mkrnion222@gmail.com; Tel.: +880-1708379269
†   These authors contributed equally to this work.

**Abstract:** Biodiversity assessment is important for evaluating community conservation status. The *haor* basin in Sylhet Division represents a transitional zone with high species availability, rare occurrences and endemism. As a result, this study aims to describe the *haor*-based freshwater fish composition, including habitat, trophic ecology, availability and conservation status. Semi-structured questionnaires were used to collect data on fish samples through focus group discussions, field surveys, and interviews with fisheries stakeholders on a monthly basis. We identified 188 morpho-species, of which 176 were finfish and 12 shellfish, distributed into 15 orders and 42 families where 29%, 42%, 15%, and 14% species were commonly available, moderately available, abundantly available, and rarely available, respectively. Cypriniformes was the dominant order in both total species and small indigenous species identified. Approximately 45.34% of species were riverine, 31.58% floodplain residents, 12.55% estuarine, 2.83% migratory, and 7.69% were exclusively hill stream residents. Carnivores and omnivores were the most dominant trophic groups. A total of 87.76% species were used as food, 12.23% as ornamental and 6.91% as sport fish. Approximately 50 species were threatened (7 critically endangered, 23 endangered and 20 vulnerable) at the national level, most of them belonging to Cypriniformes and Siluriformes. Based on endemism, 16 species were endemic of which Sygnathidae, Cobitidae, Olyridae, Cyprinidae and Balitoridae fell under the threatened category. Minimizing intense fishing efforts, banning indiscriminate fishing and destructive fishing gear, initiating fish sanctuaries and *beel* nurseries, and implementing eco-friendly modern fishing technology are suggested to conserve the threatened species. This study represents a guideline for assessing the availability and conservation of freshwater fish in the Sylhet belt and serves as a reference for decision-makers in order to allow for the sustainable exploitation of fisheries resources within an ecosystem-based framework.

**Keywords:** *haor*; biodiversity; freshwater fish conservation evaluation; threatened species

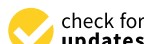



## 1. Introduction

Basic knowledge of species ordination patterns and existence images is important to accurately describe the structure and dynamics of an ecosystem [1]. Moreover, this information supports the fruitful management of natural resources and reduces possible anthropogenic effects [2,3]. Equatorial freshwater fish are highly diverse and not easily characterized by any specific features. Scientists distinguish freshwater fish into three major groups in terms of saltwater tolerance and the presumed ability to spread by overcoming maritime barriers: [4] fish that are strictly intolerant of saltwater (primary division), rarely capable of crossing narrow marine boundaries (secondary division), and representatives of marine families colonizing inland water from the sea (peripheral division). Fish

Base (http://www.fishbase.org (accessed on 7 December 2016) adapts a slightly different salient feature of freshwater and brackish water fish species into three groups: (1) entirely freshwater, (2) fresh and brackish water, and (3) fresh, brackish and seawater.

Globally, freshwater and brackish water fish species belong to 207 families and 2513 genera, of which 11,952 are strictly freshwater species [5] and 15,062 inhabit fresh and brackish waters [6]. Notably, the number assessed to be 13,000 ichthyofauna found in freshwater reservoirs covers only 1% of the earth's surface. Freshwater fish species and the stability of their existing ecosystem are seriously threatened and are the world's most endangered group of animals after amphibians [7–9].

Bangladesh is blessed with highly diverse and rich natural aquatic resources in the form of rivers, streams, estuaries, mangroves, floodplains, *haors* (seasonal wetlands), *baors* (oxbow lakes), *beels* (perennial water bodies), canals and artificial reservoir ponds. A total of 253 fish species were assessed where 104 were riverine, 113 floodplain inhabitants and 36 migratory species [10]. The freshwater fish species are not limited to freshwater; 62 species live in estuaries and numerous fish species migrate upstream from the Bay of Bengal [11]. Among the assessed fish species, 64 species were listed as threatened, comprising 25.3% of the total species assessed, while 9 species were Critically Endangered (CR), 30 species were Endangered (EN) and 25 species were Vulnerable (VU). In addition, 27 species were listed as Near Threatened (NT), 122 species as Least Concern (LC), and the remaining 40 species were considered Data Deficient (DD). No fish were found to be Extinct or Regionally Extinct [10]. In the last few decades, freshwater fish faced adverse impacts due to anthropogenic environmental degradation such as urbanization, construction of dams, diversion of water for irrigation and power generation and pollution. Unfortunately, the country's biodiversity is under threat due to the recent growing population and excessive extraction and use of natural resources [10].

Sylhet Division covers 12,558 square kilometers (with 217 *haors* covering 16,154.51 ha; 663 floodplains covering 174,824.17 ha; 3167 *beels* covering 40,946.18 ha; 116,850 ponds covering 15,129.09 ha and 20 fish sanctuaries) and is comprised of four districts (Sylhet, Sunamganj, Moulvibazar, and Hobiganj). It is the most important breeding, nursery and grazing habitats for freshwater fish species with near 0.26 million metric tons (MT) total production and a surplus of 55.85 thousand metric tons [12]. Approximately 0.152 million registered fishers are engaged in fishing and depend on natural waters for livelihood. Fishers with diverse fishing crafts and apparatuses seize a massive number of different fish species in the *haors*, rivers, and *beels* every day except during times when fishing is definitely prohibited. Indiscriminate killing, over-exploitation, use of destructive fishing gear and techniques, pollution and lack of proper management have put the fish biodiversity of Sylhet division at extreme risk. As a result, many fish have become vulnerable, endangered and critically endangered over time. The extinction of fish species at the global and local levels seriously threatens biodiversity and ecosystem balance [13]. Some research studies have been conducted on fish biodiversity in Sylhet Division, but a complete list of existing ichthyofauna with up-to-date conservation status is lacking. Therefore, it is very challenging to comprehend the present status of fish in Sylhet Division. Detailed survey work with a logical inventory of fish species is highly required to undertake necessary management for conservation of fish biodiversity of Sylhet division.

Although there are a few relevant publications on diversity and conservation status in this region, to date there is no compilation of the complete list of freshwater fish of the entire Sylhet division and information on their potential threat level. Considering this scenario, the aim of the study was to (a) prepare an updated checklist of floodplain rich freshwater fish species composition, availability status, habitat and trophic status, and national and global conservation status, and (b) propose recommendations to develop the existing conservation position of threatened fish in Bangladesh.

## 2. Materials and Methods

### 2.1. Study Area

Sylhet division lies between latitudes 23°58′ and 25°12′ north and longitudes 90°56′ and 92°30′ east. It is bordered by Meghalaya to the north, Tripura to the south, Assam to the east and Netrokona and Kishoreganj districts to the west. The study was conducted at 43 sampling sites (10 sites in fish *arat*/wholesale fish markets, 16 sites in retail fish markets, and 17 sites in fishing spots/areas) across Sylhet division (Figure 1). The study sites were selected to consider their unique geographic locations and incredible species diversity. The GPS reading of sampling spots was taken using Android GPS Test Software (version 1.6.3). A map of the sampling sites was created using ArcMap 10.7 [14].

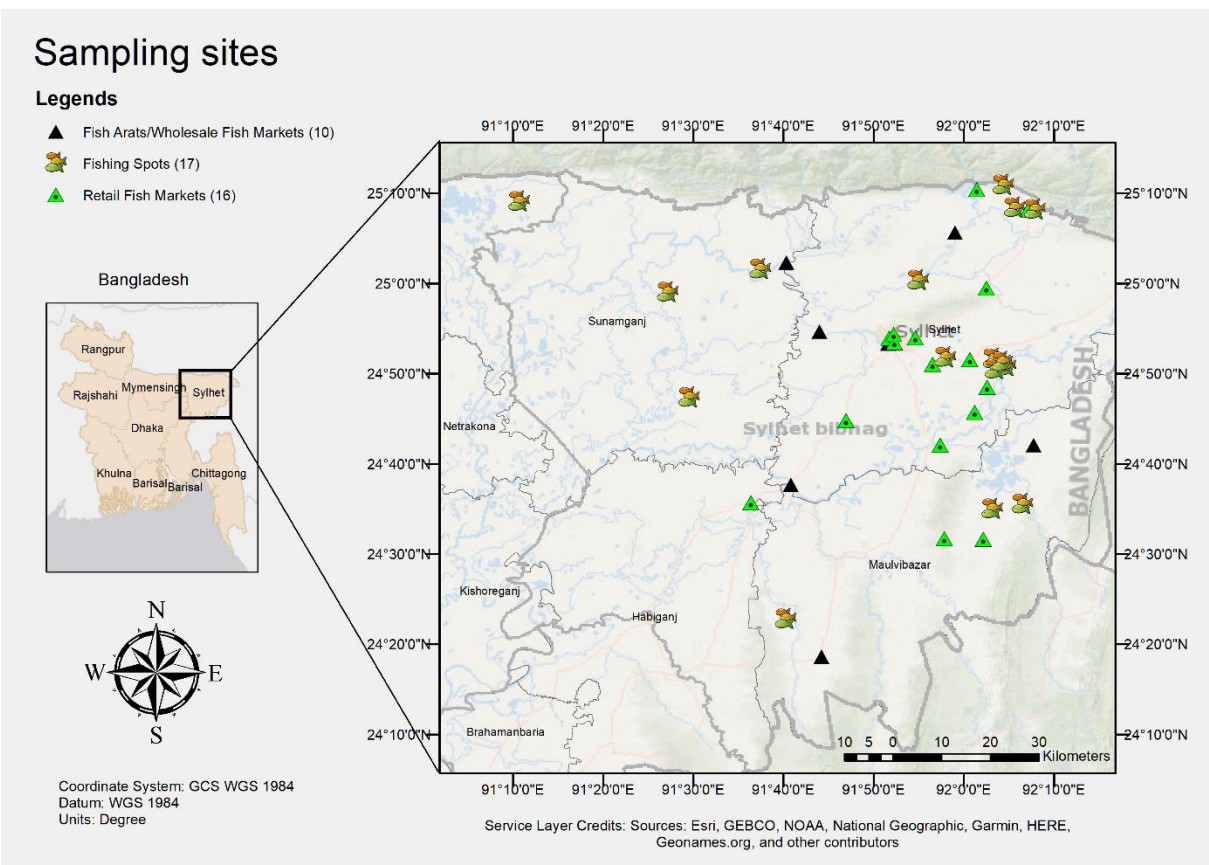

**Figure 1.** Location of the sampling sites.

### 2.2. Data Collection Framework and Species Identification

An eight-month study was conducted from September 2017 to April 2018. Focus group discussions (FGDs) with commercial fishing vessel owners, fish retailers, fish traders, locals, fishermen, sport anglers, riverbank colonials, and other people who came forward were used to obtain information regarding the feasibility of sampling existing fish species. In addition, a semi-structured questionnaire was used to conduct consultations at fish markets, fish landing centers, and fishing villages.

Fish samples were collected in both live and fresh conditions. Samples of live and fresh fish were collected directly from fishermen at fishing spots, *aratders*/wholesalers at *arat*/wholesale fish markets, and retailers at retail fish markets. For capturing live fish samples, fishing nets (seine net, gillnet, cast net, drag net, pull net, push net, and lift net) and fishing traps were used (*Doair*, *Bair*, *Chai*, *Bana*, *Bamboo pipe*, *Hogra*), and the ethical procedure approved by the 'Ethical Approval Committee of Bangladesh Agricultural University Research System (BAURES)' was followed. During sampling, photos of each fish

species were taken with a digital camera. The collected fish samples were acknowledged by examining their biometric features in accordance with published articles [6,10,11,15].

Trophic category and habitat groups were determined by following IUCN Bangladesh [10] and the Web-based related database [6]. When data were unavailable on IUCN Bangladesh and Fishbase, information was gathered from various articles published earlier [12,16–24]. As precise information on fish trophic levels is absent in the study area, the food items for each species were reviewed to explain the feeding mode according to the literature found for each taxon. The identified fish species were categorized into functional groups based on feeding mode (omnivores, carnivores, planktivores, herbivores, larvivores, and insectivores). The ecological structure was categorized into five groups, namely, riverine, hill stream, migratory, estuarine, and floodplain residents. To identify the commercial value of fish, each species was evaluated based on specific criteria for food (showed adequate growth in unit time and attained maximum size), sport (the preference of anglers), or ornamentation (based on diversified ornamental criteria *viz.* beautiful color, shape and size, banding pattern, hardiness, transparent body, calm behavior, and adhesive suckers). Endemic species were identified based on their distribution restricted to *haor* basins in Sylhet [12]. In addition, identified fish were categorized into four groups based on respondent perceptions, namely, abundantly available (AA): available in abundance all year round (frequency of occurrence: 76–100%), commonly available (CA): usually found in small numbers all year round (frequency of occurrence: 51–75%), moderately available (MA): rarely found in the study area (frequency of occurrence: 26–50%), and rarely available (RA): found infrequently in very small numbers (frequency of occurrence: 1–25%) [4,25]. The conservation status of each species was listed in accordance with the IUCN Red List of Bangladesh [10] and Threatened Species of Global Red List [16].

### 2.3. Data Analysis

To detect the most frequent freshwater fish orders, families, trophic categories, habitat group endemicity, commercial fish value, and existing conservation status, the contribution (frequency of occurrence) of each group was assessed by following equation: $F_0 = (N/n) \times 100$, where $F_0$ is % contributor or frequency of occurrence, N is the category to be calculated (order, family, or conservation status) and n is the total number of species in each group.

## 3. Results

### 3.1. Fauna Composition

The present study on freshwater fish species in the Sylhet division revealed one hundred eighty-eight (188) species, distributed into 15 orders and 42 families, detected from 43 sampling spots. The identified fish included 176 finfish (166 indigenous and the rest 10 were exotic) followed by 12 shellfish (freshwater prawn) (Table 1).

**Table 1.** Freshwater fish species recorded in Sylhet division of Bangladesh. CA—Commonly available; MA—Moderately available; AA—Abundantly available; RA—Rarely available; FP—Flood plain; HS—Hill streams; Et—Estuarine; R—River; Mgt—Migratory; NE—Not Evaluated; DD—Data Deficient; LC—Least Concern; NT—Near Threatened; VU—Vulnerable; EN—Endangered; CR—Critically Endangered; BD—National conservation status; IUCN—Global conservation status.

| Taxon | Common Name | Local Name | Present Status | Habitat | Trophic Group | Conservation Status | |
|---|---|---|---|---|---|---|---|
| | | | | | | BD | IUCN |
| Order Perciformes | | | | | | | |
| Family Anabantidae | | | | | | | |
| *Anabas testudineus* [†] | Climbing Perch | Koi | CA | FP | I | LC | DD |

**Table 1.** *Cont.*

| Taxon | Common Name | Local Name | Present Status | Habitat | Trophic Group | Conservation Status | |
|---|---|---|---|---|---|---|---|
| | | | | | | BD | IUCN |
| Family Ambassidae | | | | | | | |
| *Pseudambassis ranga* [†, ‡] | Indian Glassy Fish | Ranga Chanda | MA | FP | C | LC | LC |
| *Chanda nama* [†, ‡] | Elongate Glass-Perchlet | Lomba Chanda | MA | FP | C | LC | LC |
| *Pseudambassis lala* [†, ‡] | High-Fin Glassy Perchlet | Lal Chanda | CA | FP | C | LC | NE |
| *Parambassis thomassi* [†, ‡] | Western Ghat Glassy Perchlet | Dhipali Chanda | MA | FP | C | NE | LC |
| *Parambassis baculis* [†, ‡] | Himalayan Glassy Perchlet | Bokul Chanda | MA | FP | C | NT | LC |
| Family Badidae | | | | | | | |
| *Badis badis* [‡] | Dwarf Chameleon Fish | Napit Koi | CA | FP, HS | C | NT | LC |
| Family Gobiidae | | | | | | | |
| *Glossogobius giuris* [†, ‡] | Tank Goby | Bele | MA | R, FP, Et | O | LC | LC |
| *Brachygobius nunus* [†] | Bumblebee Goby | Baligora | CA | Et, Mgt | C | LC | NE |
| *Gobiopterus chuno* [†] | Gobius Chuno | Chuno | CA | R, Et | C | LC | DD |
| Family Nandidae | | | | | | | |
| *Nandus nandus* [†, ‡] | Mottled Nandus | Veda | CA | FP, R | C | NT | LC |
| Family Osphronemidae | | | | | | | |
| *Colisa fasciata* [†, ‡] | Giant Gourami | Khalisha | MA | FP | O | LC | LC |
| *Trichogaster lalius* [†, ‡] | Red/Dwarf Gourami | Lal/Boicha Khalisha | CA | FP | O | LC | LC |
| *Trichogaster chuna* [†, ‡] | Sunset Gourami | Chuna Khalisha | CA | FP | O | LC | LC |
| *Trichogaster labiosus* [†, ‡] | Thick-Lipped Gourami | Thoatmota Kholisha | MA | FP | O | LC | LC |
| *Ctenops nobilis* [‡] | Indian Paradise Fish | Mohua | RA | FP | C | LC | NT |
| *Pseudosphromenus cupanus* [‡] | Spike Tail Paradise Fish | Pot Koi | MA | FP | I | LC | LC |
| Family Sciaenidae | | | | | | | |
| *Johinus coitor* [†] | Coitor Croaker | Koitor Puma | MA | Et, R, FP | C | LC | LC |
| *Otolithoides pama* [†] | Pama Croaker | Crocker Puma | MA | Et, R | P | LC | NE |
| *Panna microdon* [†] | Panna Croaker | Panna Puma | MA | Et | P | NE | LC |
| Family Cichlidae | | | | | | | |
| *Oreochromis niloticus* [*, †] | Nile Tilapia | Nilotica | MA | FP | O | NE | LC |
| Family Channidae | | | | | | | |
| *Channa striatus* [†] | Striped Snakehead | Shol | CA | FP | C | LC | LC |
| *Channa marulius* [†] | Giant Snakehead | Gajar | MA | FP | C | EN | LC |
| *Channa punctatus* [†] | Spotted Snakehead | Taki | AA | FP | C | LC | LC |
| *Channa orientalis* [†, ‡] | Walking Snakehead | Cheng | AA | FP | C | LC | NE |
| Order Siluriformes | | | | | | | |
| Family Amblycipitidae | | | | | | | |
| *Amblyceps mangois* [a, ‡] | Indian Torrent Catfish | Amu/Khudi Magur | RA | HS | C | LC | LC |
| Family Bagridae | | | | | | | |
| *Batasio batasio* [†, ‡] | Tista Batasio | Batasi | CA | R | C | NT | LC |
| *Batasio tengana* [†, ‡] | Assamese/Dwarf Catfish | Tengra Batasi | MA | R | C | EN | LC |
| *Mystus tengara* [†, ‡] | Pearl Catfish | Bujuri Tengra | CA | FP | C | LC | LC |
| *Mystus vittatus* [†, ‡] | Asian Striped Catfish | Vita Tengra | AA | FP | C | LC | LC |
| *Mystus bleekeri* [†, ‡] | Day's Mystus | Golsha Tengra | AA | R, FP | C | LC | LC |
| *Mystus cavasius* [†, ‡] | Gangetic Mystus | Kabashi Tengra | MA | R | C | NT | LC |

**Table 1.** *Cont.*

| Taxon | Common Name | Local Name | Present Status | Habitat | Trophic Group | Conservation Status | |
|---|---|---|---|---|---|---|---|
| | | | | | | BD | IUCN |
| *Rama chandramara* [†, ‡] | Asian Cory | Futki Bujuri | MA | FP | O | LC | LC |
| *Mystus keletius* [†, ‡] | Keletius Mystus | Kele Tengra | MA | R | O | NE | LC |
| *Mystus armatus* [a, †] | Kerala Mystus | Armi Tengra | MA | R | C | DD | LC |
| *Rita rita* [†, ‡] | Rita | Rita | CA | R | C | LC | EN |
| *Hemibagrus menoda* [†] | Menoda Catfish | Ram Tengra | MA | R | C | NT | LC |
| *Sperata aor* [†, $] | Long Whiskered Catfish | Guji Ayre | AA | R | C | VU | LC |
| *Sperata seenghala* [†, $] | Giant River-Catfish | Tolla/Guijja Ayre | AA | R | C | VU | LC |
| Family Chacidae | | | | | | | |
| *Chaca chaca* [‡] | Square-Head Catfish | Chaka | MA | FP | C | EN | LC |
| Family Clariidae | | | | | | | |
| *Clarias batrachus* [†] | Air Breathing Catfish | Magur | CA | FP | O | LC | LC |
| Family Erethistidae | | | | | | | |
| *Conta conta* [‡] | Conta Catfish | Konta Kutakanti | CA | R | C | NT | NE |
| Family Heteropneustidae | | | | | | | |
| *Heteropneustes fossilis* [†] | Stinging Catfish | Shing | CA | FP | O | LC | LC |
| Family Olyridae | | | | | | | |
| *Olyra longicaudata* [a, ‡] | Longtail Catfish | Vot Shingi | RA | HS | C | EN | LC |
| Family Pangasiidae | | | | | | | |
| *Pangasius pangasius* [†, $] | Yellowtail Catfish | Pangas | RA | R, Et | O | EN | LC |
| *Pangasianodon hypophthalmus* [*, †] | Thailand Catfish | Thai Pangas | AA | R | O | NE | CR |
| Family Schilbeidae | | | | | | | |
| *Ailia coila* [†, ‡] | Gangetic Ailia | Kajoli | CA | R, FP | C | LC | NT |
| *Ailia punctata* [†] | Jamuna Ailia | Bashpata | CA | R, FP | H | LC | DD |
| *Clupisoma garua* [†, $] | Garua Bacha | Ghoura Bacha | CA | R | C | EN | NE |
| *Eutropiichthys murius* [†] | Murius Vacha | Muri Bacha | CA | R | C | LC | LC |
| *Eutropiichthys vacha* [†] | Batchwa Vacha | Bacha | CA | R | C | LC | LC |
| *Neotropius atherinoides* [†] | Indian Potasi | Patasi | MA | R, FP | P | LC | LC |
| *Silonia silondia* [†, $] | Silond Catfish | Silon | MA | R, Et | C | LC | LC |
| Family Siluridae | | | | | | | |
| *Ompok bimaculatus* [†] | Butter Catfish | Boali Pabda | AA | FP | O | EN | NT |
| *Ompok pabda* [†] | Pabda Catfish | Modhu Pabda | CA | FP | O | EN | NT |
| *Ompok pabo* [†] | Pabo Catfish | Kala Pabda | MA | FP | O | CR | NT |
| *Wallago attu* [†, $] | Freshwater Shark | Boal | AA | R, Mgt | C | VU | NT |
| Family Sisoridae | | | | | | | |
| *Bagarius bagarius* [†, $] | Gangetic Goonch. | Bagair | CA | R | C | CR | NT |
| *Gagata cenia* [†] | Indian Gagata | Kawa Tengra | MA | R | O | LC | LC |
| *Gogangra viridescens* [a, †, ‡] | Huddah Nangra. | Totamukh Gang Tengra | MA | R | C | LC | LC |
| *Gagata sexualis* [a, †, ‡] | Koel Gagata | Modna Gang Tengra | MA | R, Et | C | NE | LC |
| *Glyptothorax cavia* [†, ‡] | Painted Catfish | Kaviar Pathor Chata | MA | R, FP | C | DD | LC |
| *Glyptothorax platypogonoides* [‡] | White Catfish | Sada Dag Pathor Chata | MA | R | C | NE | LC |

**Table 1.** *Cont.*

| Taxon | Common Name | Local Name | Present Status | Habitat | Trophic Group | Conservation Status | |
|---|---|---|---|---|---|---|---|
| | | | | | | BD | IUCN |
| *Glyptothorax telchitta* [†, ‡] | Copper Catfish | Telichita Pathor Chata | MA | R | C | VU | LC |
| *Hara hara* [‡] | Kosi Hara | Teen Kata Hara | AA | R, FP | C | LC | LC |
| *Hara jerdoni* [‡] | Sylhet Hara | Kutakanti | AA | R, FP | C | LC | LC |
| *Erethistes pusillus* [‡] | Giant Moth Catfish | Teen Kata Pushil | CA | R, HS | C | LC | LC |
| *Pseudolaguvia shawi* [‡] | Shawi Stone Catfish | Teen Kata Shabi | MA | R | C | DD | LC |
| *Pseudecheneis sulcata* [‡] | Sucker Throat Catfish | Vot Magur | RA | R | C | DD | LC |
| *Sisor rabdophorus* [‡] | Sisor Catfish | Sisor | RA | R | C | CR | LC |
| Family Loricariidae | | | | | | | |
| *Hypostomus Plecostomus* [†, ‡] | Armored Catfish | Sucker mouth Catfish | CA | R, Et | O | NE | NE |
| Order Anguilliformes | | | | | | | |
| Family Anguillidae | | | | | | | |
| *Anguilla bengalensis* [†] | Giant Mottled Eel | Bamosh | RA | R | C | VU | NT |
| Family Moringuidae | | | | | | | |
| *Morigua raitaborua* [‡] | Purple Spaghetti Eel | Rata baura | CA | FP | O | NE | NE |
| Family Ophichthidae | | | | | | | |
| *Pisodonophis boro* [†] | Rice-Paddy Eel | Boro Balachata | CA | Et, R | C | LC | LC |
| *Lamnostoma orientalis* [†] | Oriental Sand Eelhara | Chotku Balachata | CA | Et, R | C | NE | LC |
| *Pisodonophis cancrivorus* [†] | Long-Fin Snake Eel | Motku Balachata | CA | ET | C | LC | NE |
| Order Cyprinodontiformes | | | | | | | |
| Family Aplocheilidae | | | | | | | |
| *Aplocheilus panchax* [‡] | Blue Panchax | Teen Chokha | CA | FP | L | LC | LC |
| Order Beloniformes | | | | | | | |
| Family Belonidae | | | | | | | |
| *Xenentodon cancila* [†, ‡] | Needle Fish | Kakila | CA | FP | C | LC | NE |
| Family Adrianichthyidae | | | | | | | |
| *Oryzias melastigma* [‡] | Estuarine Rice-Fish | Kanpona | CA | Et, FP | L | LC | LC |
| Family Hemiramphidae | | | | | | | |
| *Hyporhamphus limbatus* [†] | Congaturi Halfbeak | Ekthuta | MA | R, Et | C | LC | NE |
| *Dermogenys pusillus* [†] | Wrestling Halfbeak | Gollez Ekthuti | RA | R, Et | C | LC | NE |
| Order Pleuronectiformes | | | | | | | |
| Family Soleidae | | | | | | | |
| *Pseudorhombus arsius* [‡] | Large Tooth Flounder | Kathal Pata | MA | Et | C | NE | NE |
| Order Cypriniformes | | | | | | | |
| Family Cobitidae | | | | | | | |
| *Lepidocephalichthys guntea* [†, ‡] | Guntea Loach | Gutum | AA | FP | L | LC | LC |
| *Lepidocephalus thermalis* [†, ‡] | Malabar Loach | Gormi Puiya | MA | R | C | NE | LC |
| *Canthophrys gongota* [†, ‡] | Gongota Loach | Pahari Gutum | MA | R | O | NT | LC |
| *Lepidocephalichthys irrorate* [†, ‡] | Loktak Loach | Chuccha Puiya | CA | R, FP | C | VU | LC |
| *Lepidocephalichthyes annandalei* [a, †, ‡] | Annandale Loach | Annon Puiya | MA | FP | O | VU | LC |
| *Oreonectes evezardi* [†, ‡] | Poona Loach | Kuti Puiya | MA | R | P | NE | LC |
| *Pangio pangia* [‡] | Pangia Coolie-Loach | Panga | RA | R | C | LC | LC |

**Table 1.** *Cont.*

| Taxon | Common Name | Local Name | Present Status | Habitat | Trophic Group | Conservation Status | |
|---|---|---|---|---|---|---|---|
| | | | | | | BD | IUCN |
| *Botia dario* [†,‡] | Bengal Loach | Rani Mach | AA | FP | C | EN | LC |
| *Botia lohachata* [†,‡] | Y-Loach | Lohachata Rani | RA | R | C | EN | NE |
| *Botia dayi* [†,‡] | Hora Loach | Dayo Rani | MA | R | O | EN | NE |
| Family Balitoridae | | | | | | | |
| *Schistura scaturigina* [†,‡] | Scaturigina Loach | Kathuri Puiya | MA | HS | O | EN | LC |
| *Schistura corica* [†,‡] | Corica Loach | Korica Puiya | MA | HS | O | CR | LC |
| *Acanthocobitis botia* [†,‡] | Sand Loach | Balichata | RA | FP | O | LC | LC |
| *Acanthocobitis zonalternans* [†,‡] | Dwarf Zipper Loach | Thuta Puiya | MA | FP, R | O | VU | LC |
| *Schistura sikmaiensis* [a,†,‡] | River Loach | Sikim Puiya | MA | HS | C | EN | LC |
| *Syncrossus hymenophysa* [a,†,‡] | Green Tiger Loach | Bagha Puyia | MA | HS | O | NE | LC |
| Family Cyprinidae | | | | | | | |
| *Labeo rohita* [†,$] | Rohu | Rui | AA | R, Mgt | H | LC | LC |
| *Gebelion catla* [†,$] | Catla | Catla | AA | R, Mgt | H | LC | NE |
| *Cirrhinus cirrhosus* [†,$] | Mrigal Carp | Mrigal | AA | R | O | NT | LC |
| *Labeo gonius* [†] | Kuria Labeo | Gonia | AA | R | H | NT | LC |
| *Cirrhinus reba* [†,‡] | Reba Carp | Tatkini | CA | R, FP | P | NT | LC |
| *Labeo calbasu* [†,$] | Black Rohu | Kalibaus | AA | R, FP | H | LC | LC |
| *Labeo bata* [†] | Bata Labeo | Bhangon Bata | MA | R | H | LC | LC |
| *Labeo boggut* [†] | Boggut Labeo | Ghoria | MA | R | P | VU | LC |
| *Labeo boga* [†] | Boga Labeo | Boga Bata | MA | R | H | CR | LC |
| *Labeo angra* [†] | Angra Labeo | Angra Rui | RA | R | H | VU | LC |
| *Labeo ariza* [a,†] | Ariza Labeo | Ariza Rui | RA | R | P | NT | LC |
| *Labeo dero* [†] | Kusha Labeo | Kusha Rui | RA | R | H | DD | LC |
| *Chagunius chagunio* [†] | Jerruah/Chaguni | Chaguni | RA | R | O | VU | LC |
| *Labeo nandina* [†] | Nandina Labeo | Nandia Rui | RA | R | O | CR | NT |
| *Labeo pangusia* [†] | Pangusia Labeo | Ghora Mokho Rui | CA | R | H | EN | NT |
| *Labeo dyocheilus* [†] | Brahmaputra Labeo | Ghora Maach | RA | R | H | DD | LC |
| *Tor tor* [†,‡,$] | Tor Mahseer | Mohashol | RA | R | O | CR | DD |
| *Garra gotyla* [†,‡] | Sucker Head | Ghar Poia | MA | HS | H | EN | LC |
| *Puntius sarana* [†,‡] | Olive Barb | Sarpunti | CA | FP | O | NT | LC |
| *Puntius ticto* [†,‡] | Ticto Barb | Tit Punti | AA | FP, HS | O | VU | LC |
| *Puntius sophore* [†,‡] | Pool Barb | Jat Punti | AA | FP | O | LC | LC |
| *Puntius chola* [†,‡] | Swamp Barb | Chola Punti | CA | FP | O | LC | LC |
| *Puntius guganio* [†,‡] | Glass Barb | Mola Punti | MA | FP | O | LC | LC |
| *Puntius gelius* [‡] | Golden Barb | Jelly Punti | MA | FP | O | NT | LC |
| *Pethia phutunio* [†] | Dwarb Barb | Phutoni Punti | MA | FP | C | LC | LC |
| *Oreichthys cosuatis* [†,‡] | Cosuatis Barb | Kosha Punti | RA | FP | C | EN | NE |
| *Puntius conchonius* [†,‡] | Rosy Barb | Kanchon Punti | MA | FP | O | LC | LC |
| *Puntius terio* [†] | One-Spot Barb | Teri Punti | MA | FP, HS | O | LC | LC |
| *Puntius parrah* [†] | Parrah Barb | Para Punti | CA | FP | O | NE | LC |
| *Aspidoparia jaya* [†] | Carplet | Jaya | MA | R | O | LC | NE |
| *Aspidoparia morar* [a,†] | Aspidoparia | Morari | MA | R | O | VU | NE |

**Table 1.** *Cont.*

| Taxon | Common Name | Local Name | Present Status | Habitat | Trophic Group | Conservation Status | |
|---|---|---|---|---|---|---|---|
| | | | | | | BD | IUCN |
| *Chela cachius* [†] | Silver Hatchet Chela | Kechi Chela | MA | R, FP | P | VU | LC |
| *Salmostoma phulo* [†] | Fine-Scale Razor Belly Minnow | Phul Chela | CA | R, FP | P | NT | LC |
| *Salmostoma bacaila* [†] | Large Razor Belly Minnow | Narkeli Chela | MA | R | O | LC | LC |
| *Securicula gora* [†] | Gora Chela | Ghora Chela | CA | FP | C | NT | LC |
| *Amblypharyngodon mola* [†] | Mola Carplet | Mola | AA | FP, R | P | LC | LC |
| *Amblypharyngodon microlepis* [†] | Indian Carplet | Boro Mola | MA | FP, R | P | LC | EN |
| *Osteobrama cotio* [†,‡] | Cunma Osteobrama | Dhela | MA | R, FP | O | NT | LC |
| *Danio devario* [†] | Sind Danio | Chap Chela | CA | FP | C | LC | LC |
| *Danio rerio* [†,‡] | Zebra Danio | Anju | RA | HS | O | NT | LC |
| *Rasbora daniconius* [†,‡] | Blackline Rasbora | Dankina | MA | FP | O | LC | LC |
| *Bengala elanga* [†,‡] | Bengala Barb | Elang | MA | R | O | EN | LC |
| *Esomus danricus* [†,‡] | Flying Barb | Darkina | CA | FP | O | LC | NE |
| *Barilius tileo* [a,†,‡] | Tileo Baril | Tila Borali | RA | R | O | EN | LC |
| *Barilius bendelisis* [a,†,‡] | Hamilton's Barila | Hiralu Borali | RA | R, HS | O | EN | LC |
| *Barilius barila* [a,†,‡] | Barred Baril | Borali | MA | R | C | EN | LC |
| *Barilius vagra* [a,†,‡] | Vagra Baril | Vagra Borali | RA | R, HS | O | EN | LC |
| *Barilius dogarsinghi* [a,†,‡] | Manipur Baril | Dogarsingh Borali | MA | HS, R | O | NE | VU |
| *Puntius gonionotus* [*,†,‡] | Silver Barb | Thai Sarpunti | AA | R | O | NE | LC |
| *Hypopthalmichthys molitrix* [*,†] | Silver Carp | Silver | AA | R | P | NE | NT |
| *Hypopthalmichthys nobilis* [*,†] | Bighead Carp | Bighead Carp | AA | R | P | NE | DD |
| *Cyprinus carpio* [*,†] | Common Carp | Carpio | AA | R | O | NE | VU |
| *Cyprinus carpio var. communis* [*,†] | Common Carp | Carpio | AA | R | O | NE | DD |
| *Cyprinus carpio var. specularis* [*,†] | Mirror Carp | Mirror Carp | AA | R | O | NE | DD |
| *Ctenopharyngodon Idella* [*,†] | Grass Carp | Grass Carp | AA | R | H | NE | NE |
| *Mylopharyngodon piceus* [*,†] | Black Carp | Black Carp | MA | R | C | NE | DD |
| Family Psilorhynchidae | | | | | | | |
| *Psilorhynchus balitora* [†] | Balitora Minnow | Balitora | MA | HS | P | LC | LC |
| Order Clupeiformes | | | | | | | |
| Family Clupeidae | | | | | | | |
| *Tenualosa ilisha* [†] | Hilsa Shad/River Shad | Ilish | MA | R, Mgt | P | LC | LC |
| *Gudusia chapra* [†] | Indian River Shad | Chapila | AA | FP | O | VU | LC |
| *Gonialosa manmina* [†] | Ganges River Gizzard Shad | Mukh Chukka Chapila | RA | R, Et | P | LC | LC |
| *Corica soborna* [†] | Ganges River Sprat | Kachki | CA | R, FP | P | LC | LC |
| *Ilisha melastoma* [†] | Indian Ilisha | Khorchuna | MA | Mgt, Et | P | DD | LC |
| Family Engraulidae | | | | | | | |
| *Setipinna phasa* [†] | Gangetic Hairfin Anchovy | Phasa | MA | R, Et | O | LC | LC |
| Order Rajiformes | | | | | | | |
| Family Dasyatidae | | | | | | | |

**Table 1.** *Cont.*

| Taxon | Common Name | Local Name | Present Status | Habitat | Trophic Group | Conservation Status | |
|---|---|---|---|---|---|---|---|
| | | | | | | BD | IUCN |
| *Himantura bleekeri* [†] | White Nose/Dwarf Whipray | Sankush | MA | Mgt | C | NE | VU |
| Order Mugiliformes | | | | | | | |
| Family Mugilidae | | | | | | | |
| *Rhinomugil corsula* [†,‡] | Corsula Mullet | Khorsula | MA | R, Et | O | LC | LC |
| Order Osteoglossiformes | | | | | | | |
| Family Notopteridae | | | | | | | |
| *Chitala chitala* [†,‡,$] | Clown Knife Fish | Chital | CA | R | C | EN | NT |
| *Notopterus notopterus* [†,‡] | Bronze Featherback | Foli | CA | FP | C | VU | LC |
| Order Sygnathiformes | | | | | | | |
| Family Sygnathidae | | | | | | | |
| *Microphis cuncalus* [‡] | Crocodile-Tooth Pipefish | Kona Kumirer Khil | MA | R, Et | C | VU | LC |
| *Microphis deocata* [a,‡] | Deocata Pipefish | Kota Kumirer Khil | MA | R, Et | C | VU | NT |
| Order Synbranchiformes | | | | | | | |
| Family Synbranchidae | | | | | | | |
| *Monopterus cuchia* [†] | Gangetic Mud Eel. | Kuchia | CA | FP | C | VU | VU |
| *Ophisternon bengalense* [†] | Bengal Eel | Boush | RA | Et, R | C | VU | LC |
| Family Mastacembelidae | | | | | | | |
| *Mastacembelus armatus* [†,‡] | Tire-Track Spiny Eel | Shal Baim | CA | R | C | EN | NE |
| *Macrognathus aculeatus* [†,‡] | Lesser Spiny Eel | Tara Baim | CA | R | C | NT | NE |
| *Macrognathus aral* [†,‡] | One-Stripe Spiny Eel | Holdey Dora Tara Baim | RA | FP | C | DD | LC |
| *Macrognathus pancalus* [†] | Striped spiny eel | Pakal | MA | FP | C | LC | LC |
| *Mastacembelus oatesii* [†] | Inlelake Spiny Eel | Chokra Baim | CA | FP | C | NE | EN |
| Order Tetraodontiformes | | | | | | | |
| Family Tetraodontidae | | | | | | | |
| *Tetraodon cutcutia* [‡] | Ocellated Puffer Fish | Potka | CA | FP | C | LC | LC |
| *Tetraodon nigroviridis* [‡] | Spotted Green Pufferfish | Bish Potka | MA | FP | C | NE | NE |
| Order Decapoda | | | | | | | |
| Family Palaemonidae | | | | | | | |
| *Macrobrachium rosenbergii* [†] | Giant Freshwater Prawn | Golda Chingri | CA | R. Et | O | LC | LC |
| *Macrobrachium malcolmsonii* [†] | Monsoon River Prawn | Chatka Chingri | CA | R, Et | O | LC | LC |
| *Macrobrachium rude* [†] | Hairy River Prawn | Paitta Chingri | CA | R, Et | O | LC | LC |
| *Nematopalaemon tenuipes* [†] | Spider Prawn | Gura Chingri | CA | Et | O | DD | NE |
| *Macrobrachium dolichodactylus* [†] | Ghoda River Prawn | Gada Icha | CA | R, Et | O | LC | NE |
| *Macrobrachium Idella* [†] | Slender River Prawn | Chikna Icha | MA | R | O | DD | LC |
| *Macrobrachium villosimanus* [†] | Dimua River Prawn | Dimua Icha | MA | R | O | LC | LC |
| *Macrobrachium lamarrei* [†] | Kuncho River Prawn | Kunchu Icha | MA | R, Et | O | LC | LC |
| *Macrobrachium birmanicum* [†] | Birma River Prawn | Thengua Icha | MA | R, Et | O | LC | LC |
| *Macrobrachium lar* [†] | Tahitian Prawn | Chora Icha | MA | R, Et | O | DD | LC |
| *Macrobrachium dayanus* [†] | Kaira River Prawn | Ghoda Icha/Beel Icha | MA | R | O | LC | NE |
| *Macrobrachium equidens* [†] | Rough River Prawn | Goda Icha | MA | R, Et | O | DD | LC |

* Exotic species, [a] Endemic species, [†] Food fish, [‡] Ornamental fish, [$] Sportfish.

Table 2 summarizes a comparison of recorded fish species and their presence (%) to the national and global levels. Compared with nationwide levels, Synbranchiformes showed the highest prevalence (116.67%) followed by Osteoglossiformes (100%), Cyprinodontiformes (100%), Siluriformes (83.64%) and Cypriniformes (79.35%). Compared with worldwide levels, the highest presence was occupied by Anguilliformes (83.33%), followed by Tetraodontiformes (16.67%), Mugiliformes (12.5%) and Pleuronectiformes (10%). The rest were less than 10% compared with global prevalence.

**Table 2.** Status of freshwater fish species in Sylhet division compared with national (BD) and global level.

| Class | Order | Number of Freshwater Fish Species | | | Species Presence (%) Compared to National and Global Levels | |
|---|---|---|---|---|---|---|
| | | **National** | **Global** | **Present Study** | **National** | **Global** |
| Actinopterygii | Cypriniformes | 92 [b] | 2662 [a] | 73 | 79.35 | 2.74 |
| | Siluriformes | 55 [b] | 2280 [a] | 46 | 83.64 | 2.02 |
| | Perciformes | 56 [b] | 1922 [a] | 25 | 44.64 | 1.3 |
| | Clupeiformes | 17 [b] | 79 [a] | 6 | 35.29 | 7.59 |
| | Anguilliformes | 8 [h] | 6 [a] | 5 | 62.5 | 83.33 |
| | Beloniformes | 6 [b] | 98 [a] | 4 | 66.67 | 4.08 |
| | Sygnathiformes | 3 [b] | 81 [d] | 2 | 66.67 | 2.47 |
| | Synbranchiformes | 6 [h] | 94 [a] | 7 | 116.67 | 7.45 |
| | Tetraodontiformes | 3 [h] | 12 [a] | 2 | 66.67 | 16.67 |
| | Osteoglossiformes | 2 [b] | 244 [a] | 2 | 100 | 0.82 |
| | Cyprinodontiformes | 1 [b] | 996 [a] | 1 | 100 | 0.1 |
| | Pleuronectiformes | 4 [b] | 10 [a] | 1 | 25 | 10 |
| | Mugiliformes | 6 [b] | 8 [e] | 1 | 16.67 | 12.5 |
| Elasmobranchii | Rajiformes | 5 [c] | 59 [d] | 1 | 20 | 1.69 |
| Malacostraca | Decapoda | 24 [f] | 800 [g] | 12 | 50 | 1.5 |

[a] = Nelson et al. [26]; [b] = IUCN Bangladesh [10]; [c] = Froese and Pauly [6]; [d] = Eschmeyer [27]; [e] = Nelson [5]; [f] = Ahmed et al. [28]; [g] = De Grave et al. [29]; [h] = Hossain and Wahab [30].

Cypriniformes was the dominant order (39.04%) followed by Siluriformes (24.6%) and Perciformes (13.37%). The rest of the faunal orders contributed approximately 23.53% of the total species found (Table 3). Cyprinidae was the leading family, accounting for 29.95% (56 species) of all families identified, followed by Bagridae with 6.95% (13 species), Sisoridae with 6.95% (13 species), Palaemonidae with 6.42% (12 species), and Cobitidae with 5.35% (10 species).

The fish recorded in this study were categorized into 17 major groups: perch, snakeheads, catfish, eels, tooth carp, needlefish, flatfish, barbs and minnows, carp, clupeids, stingrays, mullets, feather backs, pipefish, pufferfish, prawns, and loach (Figure 2). Catfish contributed the most, accounting for 25% of the total groups, followed by barbs and minnows (17%), carp (13%), perch fish (11%), loach (8%), prawns (6%) and eels (6%). The remaining groups contributed a considerably smaller percentage (14%). Small indigenous species (SIS) comprised 140 species distributed into 12 orders, of which Cypriniformes (56 species), Siluriformes (39 species) and Perciformes (22 species) were dominant, representing approximately 83.57% (Figure 3).

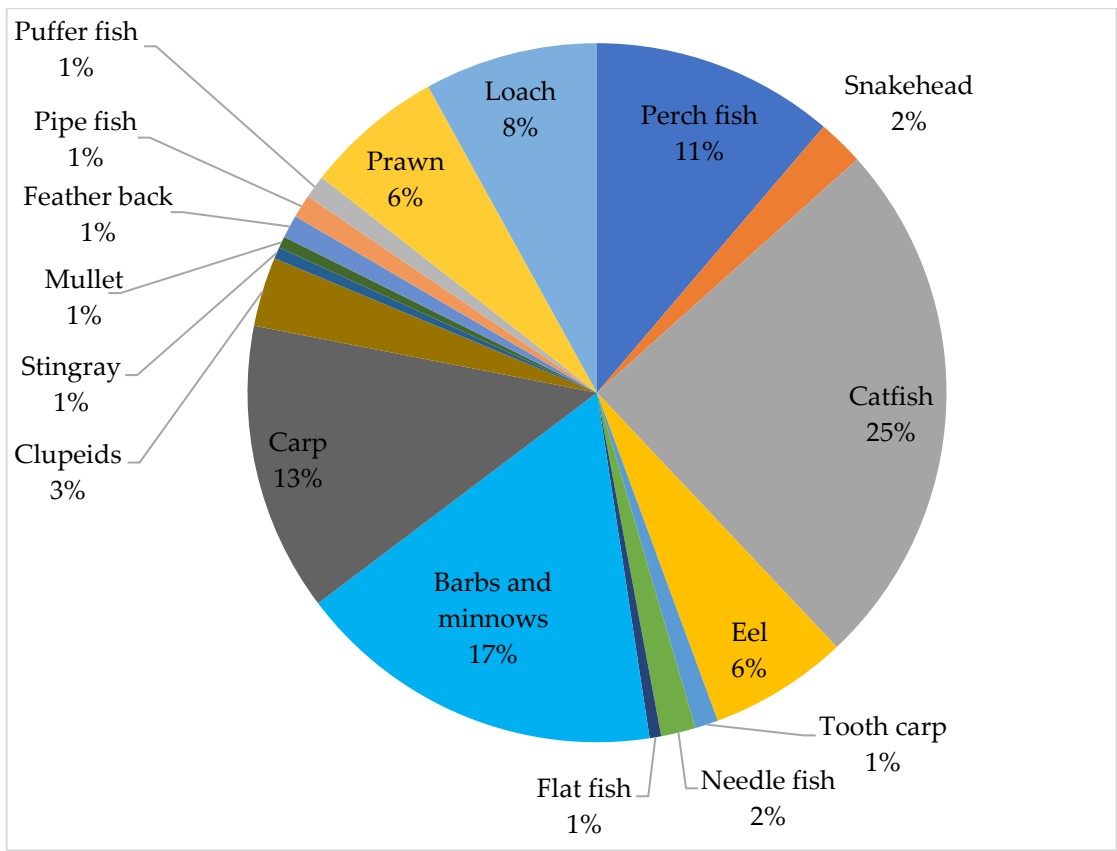

**Figure 2.** Major groups of freshwater fish in Sylhet division.

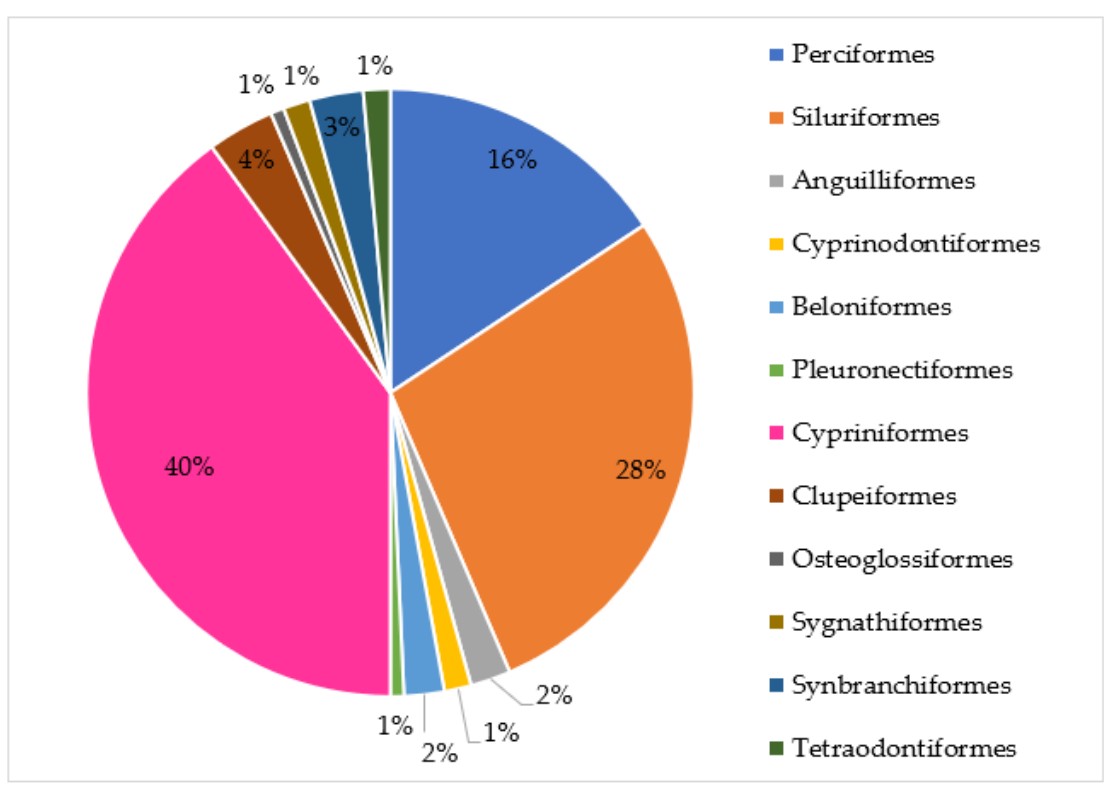

**Figure 3.** Composition of small indigenous fish species (based on order) from the study area.

**Table 3.** Number and frequency of occurrence (FO) of orders and families of fish in Sylhet division, Bangladesh.

| Taxon | N | FO (%) | Taxon | N | FO (%) |
|---|---|---|---|---|---|
| Order Perciformes | 25 | 13.37 | Order Clupeiformes | 6 | 3.21 |
| Family Anabantidae | 1 | 0.53 | Family Clupeidae | 5 | 2.67 |
| Family Ambassidae | 5 | 2.67 | Family Engraulidae | 1 | 0.53 |
| Family Badidae | 1 | 0.53 | Order Cypriniformes | 73 | 39.04 |
| Family Nandidae | 1 | 0.53 | Family Cobitidae | 10 | 5.35 |
| Family Osphronemidae | 6 | 3.21 | Family Balitoridae | 6 | 3.21 |
| Family Gobiidae | 3 | 1.60 | Family Cyprinidae | 56 | 29.95 |
| Family Sciaenidae | 3 | 1.60 | Family Psilorhynchidae | 1 | 0.53 |
| Family Cichlidae | 1 | 0.53 | Order Rajiformes | 1 | 0.53 |
| Family Channidae | 4 | 2.14 | Family Dasyatidae | 1 | 0.53 |
| Order Siluriformes | 46 | 24.60 | Order Osteoglossiformes | 2 | 1.07 |
| Family Amblycipitidae | 1 | 0.53 | Family Notopteridae | 2 | 1.07 |
| Family Bagridae | 13 | 6.95 | Order Mugiliformes | 1 | 0.53 |
| Family Chacidae | 1 | 0.53 | Family Mugilidae | 1 | 0.53 |
| Family Clariidae | 1 | 0.53 | Order Sygnathiformes | 2 | 1.07 |
| Family Erethistidae | 1 | 0.53 | Family Sygnathidae | 2 | 1.07 |
| Family Heteropneustidae | 1 | 0.53 | Order Anguilliformes | 5 | 2.67 |
| Family Olyridae | 1 | 0.53 | Family Anguillidae | 1 | 0.53 |
| Family Pangasiidae | 2 | 1.07 | Family Moringuidae | 1 | 0.53 |
| Family Schilbeidae | 7 | 3.74 | Family Ophichthidae | 3 | 1.60 |
| Family Siluridae | 4 | 2.14 | Order Synbranchiformes | 7 | 3.74 |
| Family Sisoridae | 13 | 6.95 | Family Synbranchidae | 2 | 1.07 |
| Family Loricariidae | 1 | 0.53 | Family Mastacembelidae | 5 | 2.67 |
| Order Beloniformes | 4 | 2.18 | Order Tetraodontiformes | 2 | 1.07 |
| Family Belonidae | 1 | 0.53 | Family Tetraodontidae | 2 | 1.07 |
| Family Hemiramphidae | 2 | 1.07 | Order Cyprinodontiformes | 1 | 0.53 |
| Family Adrianichthyidae | 1 | 0.53 | Family Aplocheilidae | 1 | 0.53 |
| Order Pleuronectiformes | 1 | 0.53 | Order Decapoda | 12 | 6.42 |
| Family Soleidae | 1 | 0.53 | Family Palaemonidae | 12 | 6.42 |

### 3.2. Habitat Status and Trophic Ecology of Fish Fauna

Figure 4 reveals that 45.34% of the species were riverine, 31.58% floodplain residents, 12.55% estuarine, 2.83% migratory (traveled to floodplains and other habitats for feeding and spawning during monsoon) and 7.69% were exclusively hill stream inhabitants. Figure 4 also demonstrates that carnivorous and omnivorous species were the leading trophic groups, accounting for 43.62% and 37.23%, respectively, followed by planktivorous with 9.57%, herbivorous with 6.91%, larvivorous with 1.6%, and insectivorous with 1.06%.

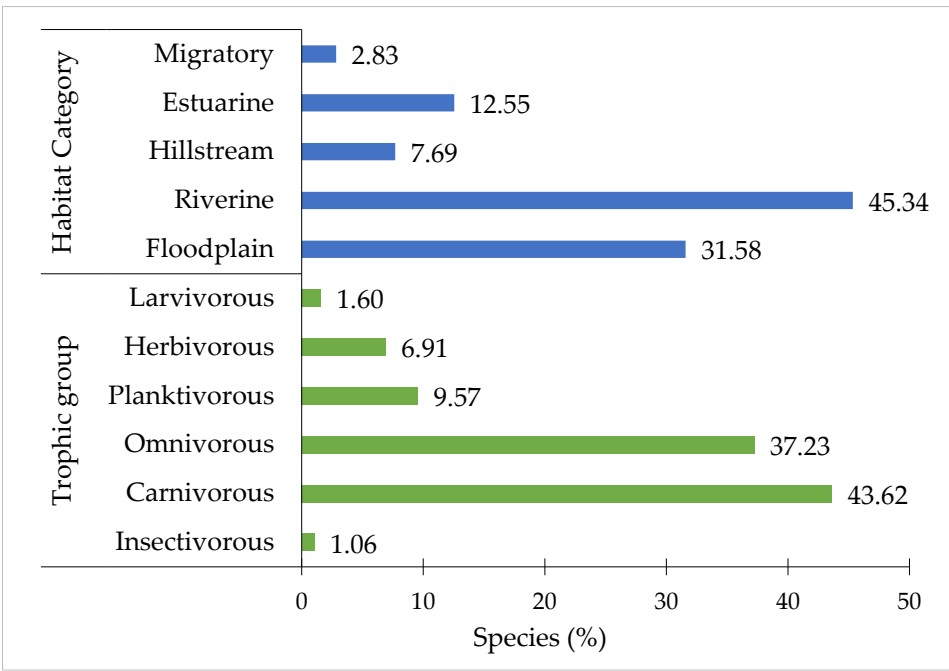

**Figure 4.** Habitat structure and trophic groups of freshwater fish recorded in Sylhet division.

### 3.3. Endemic Status of Fish

Figure 5 summarizes the species-rich group of endemic species from the study area. A total of sixteen (16) endemic species were identified, distributed across three orders and eight families, accounting for 8% of the total species found. Cyprinidae was the leading endemic species-rich family (seven species—*Labeo ariza*; *Aspidoparia morar*; *Barilius tileo*; *Barilius bendelisis*; *B. barila*; *B. vagra*; *B. dogarsinghi*) with 44% of total endemic species found, followed by Balitoridae with 13% (two species—*Schistura sikmaiensis*; *Syncrossus hymenophysa*) and Sisoridae with 13% (two species—*Gogangra viridescens*; *Gagata sexualis*). The rest contributed 30% (one species of each family) to the total endemic species found.

### 3.4. Commercial Utilization Status of Fish

The utilization of commercial fish revealed that 87.76% (165 species) were consumed as food, with 43.03% also having ornamental value (71 sp.). Approximately 12.23% (23 species) were only of ornamental value, while 6.91% (13 species) were considered sport fish with food value. *Tor tor* and *Chitala chitala* were considered to be food, ornamental, and sport fish (Table 1). All species belonging to Decapoda, Rajiformes, and Clupeiformes were found to have food value, while Tetraodontiformes, Pleuronectiformes, Cyprinodontiformes, and Sygnathiformes had ornamental value. Cypriniformes, Osteoglossiformes, and Siluriformes were shown to have food, ornamental, and sport fish value, with food value being more prevalent (62.28%) compared to the other groups (40% and 52.17% respectively) (Figure 6).

### 3.5. Present Status of Identified Fish Species

The current study revealed that approximately 42% of fish were moderately available, followed by 29% commonly available, 15% abundantly available and 14% rarely available (Figure 7a). Figure 7b summarizes the present status of species according to order. The results showed that moderately available, rarely available and abundantly available species were highest in Cypriniformes (39.24%, 57.69% and 58.62% respectively) whereas commonly available species were highest in Siluriformes (27.78%).

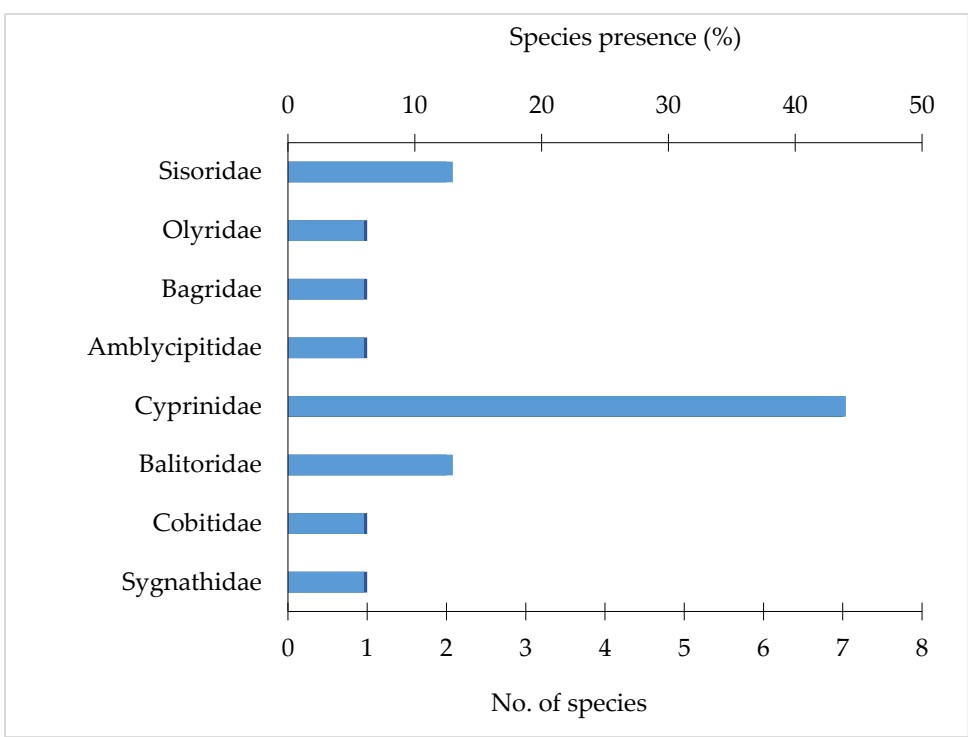

**Figure 5.** Diagrammatic representation of families rich in endemic freshwater species in Sylhet division.

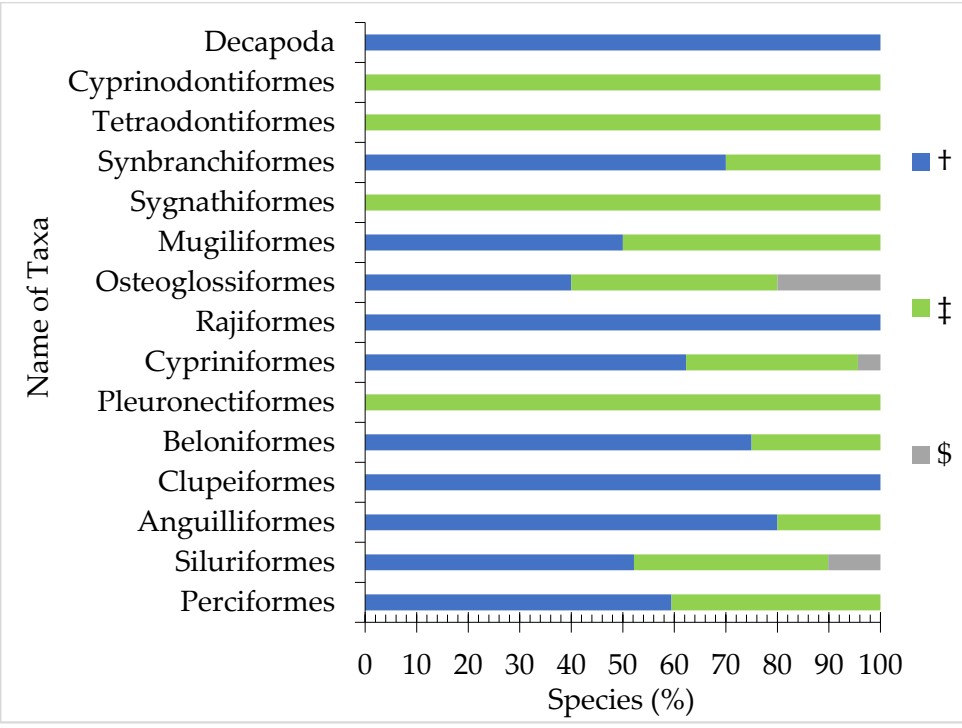

**Figure 6.** Relative frequency (%) of commercial utilization of fish. † Food fish, ‡ Ornamental fish, $ Sport fish.

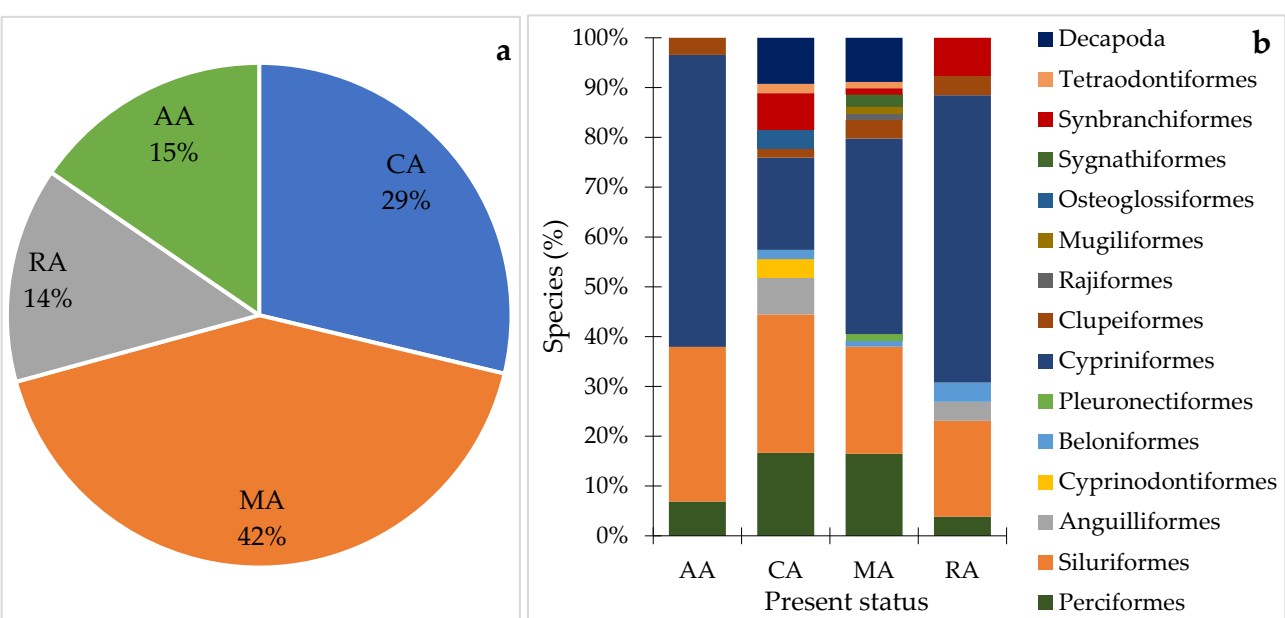

**Figure 7.** Present status of identified fish (**a**) and frequency of occurrence (%) based on order (**b**). AA = Abundantly available; CA = Commonly available; MA = Moderately available; RA = Rarely available.

*3.6. Conservation Status*

A total of 161 species (86.1%) and 157 species (83.95%) were evaluated in the national list [10] and global list, respectively. Of these recorded species, 50 species (26.74%) were considered threatened at a national level (CR—7 sp.; EN—23 sp. and VU—20 sp.) and eight species (4.27%) at a global level (CR—1 sp.; EN—3 sp.; VU—4 sp.). The majority of recorded species were listed as LC both in the present study (42.55%) and globally (70.21%). Out of 188 species, 19 species were listed as NT, and 11 as DD, differing from their global category. We obtained a substantially large number of species (26 species in national status and 31 species in global) that have not been assessed in IUCN evaluation (Table 4).

**Table 4.** Number and frequency of occurrence of threat categories of fish recorded in Sylhet division, Bangladesh.

| Conservation Status | National | | Global | |
|---|---|---|---|---|
| | N | % | N | % |
| CR—Critically endangered | 7 | 3.72 | 1 | 0.53 |
| EN—Endangered | 23 | 12.23 | 3 | 1.60 |
| VU—Vulnerable | 20 | 10.64 | 4 | 2.13 |
| NT—Near threatened | 19 | 10.11 | 13 | 6.91 |
| LC—Least concern | 80 | 42.55 | 132 | 70.21 |
| DD—Data Deficient | 12 | 6.38 | 8 | 4.26 |
| NE—Not Evaluated | 27 | 14.36 | 27 | 14.36 |

Most species belonging to Cypriniformes (CR 57.14%, EN 56.52%, VU 45%) and Siluriformes (CR 42.86%, EN 30.43%, VU 20%) were listed as threatened in the present study (Figure 8a). Based on endemism, species belonging to Sygnathidae, Cobitidae and Olyridae were listed as VU and EN, respectively. Approximately 50% and 57.14% species from Balitoridae and Cyprinidae, respectively, were in the EN category (Figure 8b).

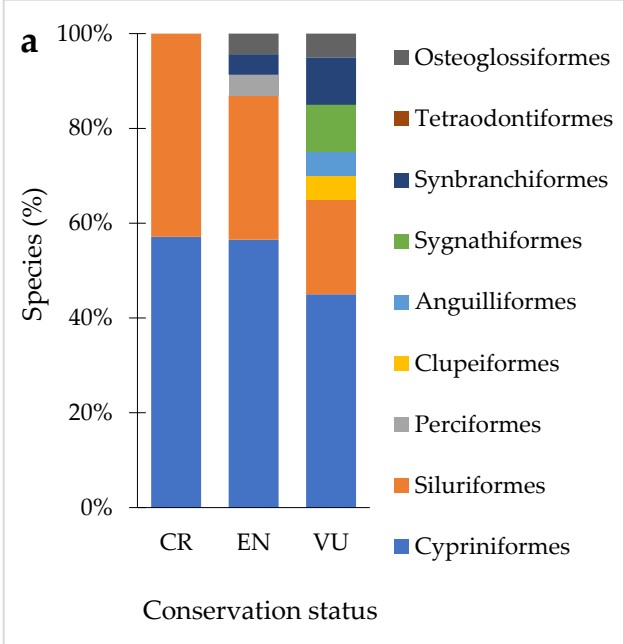
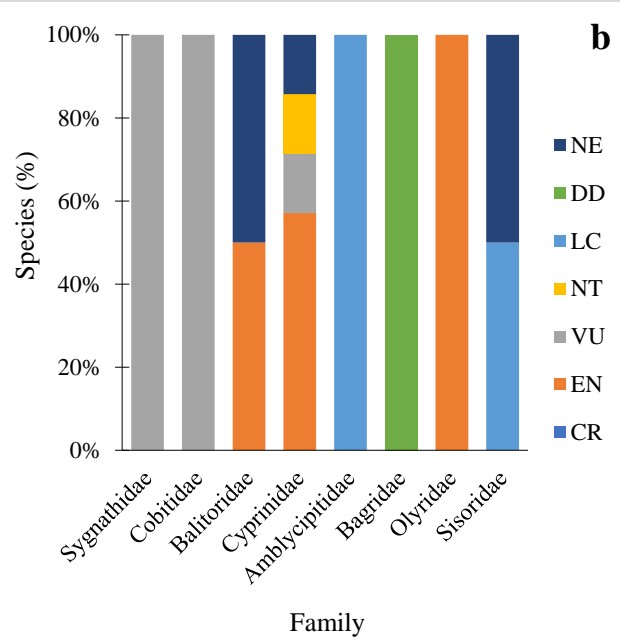

**Figure 8.** Conservation status of fish: (**a**) all combined fish; (**b**) endemic species. CR—Critically endangered; EN—Endangered; VU—Vulnerable; NT—Near threatened; LC—Least concern; DD—Data Deficient; NE—Not Evaluated.

## 4. Discussion

### 4.1. Fauna Composition

Freshwater fish species are represented in all areas at the national level, which is one of the richest diversity of fish fauna with a rich variety of morpho-species. Approximately 253 fish species have been identified from freshwater in Bangladesh [10]. Rahman [11] recorded 265 species comprising 154 genera and 55 families and Hossain et al. [31] obtained 293 species of freshwater fish including 13 orders and 61 families in Bangladesh, both of which were higher than the present study. The most species-rich orders that covered all types of water bodies and ecosystems in the Sylhet division are similar to those found in national and global surveys, and there were six orders for which the proportion of freshwater species in the Sylhet division was 67.87% of national prevalence and 2.16% of global prevalence (Table 2). This indicates that freshwater fish in the Sylhet Belt play a significant role in the national and global freshwater fish species bank. The fish belonging to the Cyprinidae family in the present study were the most dominant, a common feature of the fish community similar to Asian rivers [32–34]. Nonetheless, little research has been conducted on the fish biodiversity and conservation status in the greater Sylhet region. For instance, lower freshwater fish diversity has been reported in *haor* and wetland ecosystems at the district level within the Sylhet division [31–35], which may be attributed to extremely insufficient sampling areas. Ray and Grassle [36] noted that hydrographic conditions, climatic patterns, and habitat are influential factors that drive the number of species. Moreover, the observed image of species diversity during the study period may also be influenced by the sampling strategies and efforts used [37].

The number of small indigenous fish (140 species) listed in the entire Sylhet division was similar to the number found in Bangladesh [38]. The leading SIS orders in the study area were Cypriniformes, Perciformes, and Siluriformes, similar to observations from the Gorai River [39] and the southern coastal waters of Bangladesh [40]. Geographically, the connection of freshwater habitats such as rivers, floodplains, and diverse landscape areas maintains continuity to facilitate the movement of SIS [33].

Exotic species are extensively cultured in Bangladesh but are also found in open aquatic habitats, most likely due to escape from adjacent ponds and periodic water bodies

during flash floods [40]. Our synthesis found 176 freshwater fish, of which 166 were indigenous and the remaining 10 exotic. Mukul et al. [40] found 16 exotic freshwater fish species in Bangladesh. Galib and Mohsin [41] listed 92 varieties of exotic fish in Bangladesh, of which 16 were cultured fish species and the rest were ornamental fish species. Islam and Hossain [42] estimated a total of 57 fish species in Dekar *haor*, of which 52 were native and 5 were exotic. Suravi et al. [43] highlighted 51 species in Dekar *haor*, of which 47 were indigenous and 4 were exotic. Sayeed et al. [44] and Mahalder and Mustafa [45] reported seven and nine exotic freshwater fish species in the Hakaluki *haor* in Moulovibazar and Sunamganj regions, respectively, within CBRMP's working area. Exotic species may harm the food web and breeding ground for native species, leading to the depletion of natural reserves of endemic species and the extinction of some native species [46]. Therefore, greater emphasis should be placed on preventing the potential negative effects of exotic species on native stocks.

### 4.2. Habitat and Trophic Ecology of Fish

Floodplain-dwelling fish take shelter in nearby perennial water bodies, such as rivers and deep *beels*, when the floodplain's water level decreases during the dry season, which complicates their classification. According to IUCN Bangladesh [10], floodplain species dominate. Similarly to the present study, Pandit et al. [47] found that 54.9% of the species of the Dhanu river and surrounding wetlands in Kishoreganj were *beel* residents and 45.1% were riverine residents. Based on their habitat preference, freshwater fish spend the majority of their lives in rivers and/or floodplains, where they tend to live for much longer than in other types of freshwater environments. Feeding is the leading activity throughout the entire life cycle of fish [46]. The current trophic structure study revealed the dominance of carnivores and omnivore fishes, which corresponded to previous findings [14,17,20,48]. Sarker et al. [49] observed similar composition of feeding types in the Western Ghats and Ganges river in India. Nevertheless, several fish species had multiple trophic levels depending on their ecological resources or prey availability [50], which was similar to the current findings.

### 4.3. Endemic Status of Fish

An endemic fish is defined as a fish species localized in a particular area or country where it originated. As no previous research has been conducted on the endemic fish species of *haor* in Sylhet division, the present findings can be compared to those of neighboring countries. However, Dey et al. [20] at the Ganges river in India and De Silva et al. [51] in South-East Asia reported similar patterns. Higher endemicity was reported by [52,53] in Assam and the neighboring North-Eastern states of India (48 and 33 species respectively). The presence of a high number of endemic species in the aforementioned Indian states could be attributed to hilly terrain and perennial wetlands. Therefore, a comprehensive study of the aforementioned fish endemicity variations in the Sylhet region is required.

### 4.4. Commercial Utilization Status of Fish

Nature provides a wide variety of fish species used as food, which differ in shape and taxonomic group [54]. According to [19,20], in West Bengal, India, food fish were dominant over ornamental fish, resulting in a reverse pattern. We identified 94 native fish out of 188 fish species as ornamental fish. Some of them are already used as ornamental fish, while others could be used based on their diverse ornamental criteria *viz.* beautiful color, shape and size, banding pattern, hardiness, transparent body, calm behavior, and adhesive suckers [24]. For instance, a lower number of potential indigenous freshwater fish and non-fish species (31 spp.) were identified as having ornamental value in Bangladesh [55]. It is difficult to list the commercial utilization of fish due to the lack of preceding evidence on fish diversity, but the current study could be considered baseline information for upcoming commercial utilization status investigation.

### 4.5. Present Status of Identified Fish Species

The current status of freshwater fish is consistent with that of the fish communities in various wetland ecosystems in the *haor*-based, floodplain-rich Sylhet division [35,44,46,56,57]. Observing the current status of fish, it is possible that a large proportion of the fish fauna under Cypriniformes and Siluriformes classified as rarely available in that region will disappear in the near future.

### 4.6. Conservation Status

IUCN classification is widely used for evaluating the conservation status of fish around the world. However, due to the lack of data on the regional list, it was impossible to assess the species' regional, national and global conservation status to validate a similar pattern. According to IUCN Bangladesh [10], 25.3% of species are listed under a threatened category. We highlighted that 26.74% of our species found were considered to be threatened at the national level. This result reflects the findings reported by [44,58,59] at different *haors* and other water bodies in the Sylhet division. The threatened species composition recorded in the present study was lower than in earlier reports from Sylhet Sadar and Dekhar *haor* in Sunamganj District [60,61]. According to Hossain and Wahab [30] and IUCN Bangladesh [10], most of the fish belonging to Cypriniformes and Siluriformes faced significant threat, and the majority of them were categorized as either endangered or critically endangered over the last 10 years, which was similar to the present findings. Species listed as critically endangered experienced at least an 80% population decline over the past 10 years or three generations, indicating a significant threat to extinction in near future in the Sylhet region.

*Pangasianodon hypophthalmus*, a Critically Endangered species at the global level, infiltrated Bangladesh in the early 1990s and is now an intensively cultured species that frequently occupies freshwater bodies on a large scale [62]. Chowdhury et al. [63] reported that *Tor tor, Pangasius pangasius* and *Anguilla bengalensis* were extinct at the regional level. Islam et al. [34] and Chakraborty and Mirza [64] demonstrated that two species, *viz. Tor tor and Labeo nandina*, were extinct from the Juri River in Sylhet and the Someswari River in Netrokona respectively, compared to 10–20 years previously. *Channa barca* was found to be regionally extinct from the survey area during the study period. Furthermore, species identified as Critically Endangered at the national level, such as *Ompok pabo, Bagarius bagarius, Sisor rabdophorus, Schistura corica, Labeo boga, Labeo nandina,* and *Tor tor* were captured in limited numbers but should be prioritized for conservation owing to their population decline. According to IUCN Bangladesh [10], *Wallago attu* was listed as Vulnerable and *Notopterus notopterus, Canna marulius, Mystus armatus* as Endangered; however, these species have recently moved into the Least Concern and Endangered categories, respectively, due to signs of population growth and are now abundant in the *haor* [43].

We synthesized 27 species considered Not Evaluated, of which 10 were exotic and the remaining 17 were indigenous. Due to a lack of indigenous entities, these invasive species were not included in IUCN Bangladesh [10]. Four species, *Tetraodon nigroviridis, Pseudorhombus arsius, Hypostomus Plecostomus,* and *Morigua raitaborua*, were used as ornamental fish at national and global levels, with *Hypostomus Plecostomus* having both food and ornamental value. Thirteen species listed as Not Evaluated at national levels with only food value, including *Mastacembelus oatesii, Himantura bleekeri, Puntius parrah, Lamnostoma orientalis, Panna microdon, Barilius dogarsinghi, Syncrossus hymenophysa, Oreonectes evezardi, Lepidocephalus thermalis, Parambassis thomassi, Mystus keletius*, and *Gagata sexualis*, were used as food for local inhabitants as well as recreational purposes; *Glyptothorax platypogonoides* was considered an ornamental fish among the non-evaluated fish [65]. Additionally, a significant number of species listed as Not Threatened and Data Deficient by IUCN Bangladesh [10] require further investigation at the regional and national levels in subsequent assessments. Assessment should be prioritized at the regional level for conservation and justifying the current status of endemic species.

Local environmental knowledge, participatory planning with fisheries stakeholders, and the adoption of sustainable fisheries management practices could be the first steps in eradicating the decline in threatened species diversity and availability [66]. Fishing during the breeding and spawning seasons, indiscriminate harvesting of fish larvae and fingerlings, and the use of harmful fishing gear and crafts must be prohibited immediately. Declaring some parts of or the whole of a *haor* as a "fish sanctuary", and the concept of a "*beel* nursery", could be effective steps toward conserving endangered and vulnerable species in Sylhet division. Breeding and nursery grounds, migration routes and hotspots of fish biodiversity in the *haor* region must be designated as nature reserves and delineated by a demarcation line, with fishing strictly prohibited and navigation temporarily stopped during the breeding season. Fishing limitations by completely drying out water bodies and regular dredging of silted water are required to facilitate fish habitat, breeding, nursing, maturation, and relocation. Eco-friendly fishing technologies for the monitoring, controlling and surveillance of protected areas and threatened species, selection of fishing gear and crafts, and development of a digital fishing calendar for effective banning periods and catch restrictions should be initiated to conserve the threatened fish species [67]. In addition, a live fish gene bank could be an effective way to conserve threatened species. However, the most important aspect of conserving the threatened fish of *haor* in Sylhet division is to raise awareness among the stakeholders through effective communication, collaboration, and education. Furthermore, financial support from the government and donor agencies is crucial for further research and monitoring, along with raising awareness among fishers regarding the importance of conserving fish diversity in the *haor* areas. In short, since fish and fisheries in this region support the livelihoods of thousands of marginalized poor, particularly fishers, the government should adopt a long-term conservation strategy to ensure sustainable production in the *haor* region in Sylhet division.

## 5. Conclusions

The number of species documented during the study is a good indication of rich biodiversity in the Sylhet division. Fish species belonging to Cypriniformes and Siluriformes face significant threat levels. In addition, species that are critically endangered, endangered and vulnerable at national and global levels are intensively being cultured and a live gene bank is being established to conserve threatened species. Furthermore, the findings of this study could serve as an important benchmark for assessing biodiversity and fish conservation in the *haor* region. Notably, a large number of fish might have been excluded from evaluation due to insufficient scientific research. The threatened fish recorded in the Sylhet division indicate an alarming threat to fish conservation. Community and ecosystem-based co-management programs that promote the conservation of biodiversity and social protection schemes can be very effective to conserve fish diversity. However, fish sample collection from fishers, rather than direct sampling, limited sample size, and taxonomic identification through barcoding due to lack of funding were the major drawbacks of the present study. In addition, current status and threat level were recorded based on fishers' perceptions via a survey and researcher observation. In order to conserve fish biodiversity in this area, a thorough study is needed on species composition and assemblages, along with species' taxonomic identification, life history, geographic range, ecology and reproductive biology.

**Author Contributions:** Conceptualization, F.C.S. and M.K.R.; methodology, F.C.S. and M.K.R.; formal analysis, M.K.R. and F.C.S.; investigation, F.C.S. and M.K.R.; resources, F.C.S. and M.K.R.; data curation, M.K.R.; writing—original draft preparation, M.K.R. and F.C.S.; writing—review and editing, F.C.S., M.K.R., A.S., M.A.S. and A.K.M.N.A.; visualization, M.K.R., M.A.S. and A.S.; supervision, F.C.S., M.K.R. and A.K.M.N.A. All authors have read and agreed to the published version of the manuscript.

**Funding:** This research received no external funding.

**Institutional Review Board Statement:** The study was approved by the 'Ethical Approval Committee of Bangladesh Agricultural University Research System (BAURES)' (protocol code: 502/BAURES/ESRS/FISH/20 and date of approval: 20 March 2022).

**Informed Consent Statement:** Not applicable.

**Data Availability Statement:** The data presented in this study are available on request from the corresponding author.

**Acknowledgments:** The authors are grateful to the Department of Fisheries of the People's Republic of Bangladesh for providing financial support for data collection and to stakeholders in the fishing community across all of Sylhet division for their support in sampling and identifying fish samples.

**Conflicts of Interest:** The authors declare that they have no competing financial interests or personal relationships that may affect the work reported in this paper.

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
