# Peer review of "Haor-Based Floodplain-Rich Freshwater Ichthyofauna in Sylhet Division, Bangladesh: Species Availability, Diversity, and Conservation Perspectives"

_conservation, doi:10.3390/conservation2040042_

Round 1

Reviewer 1 Report

Brief summary:

The paper describes the fish composition and conservation status in the haor based freshwater in Bangladesh. It mainly contributes to providing a list of freshwater species in the Sylhet division and the species potential threat level, based on the conducted survey. The paper has valuable data based on empirical level. However, work still needs to be done in terms of the focus of the paper and shifting weight from just presenting to making a relevant story out of the data. Below are some general and specific comments for improving the paper.

General concepts comments:

The paper reads more like a technical report rather than a scientific article, since the focus is on enumerating results and numbers instead of highlighting key findings. In general, too many numbers in Tables as well as percentages are mentioned in the text without clearly mentioning their relevance. Where is the story, why would it be interesting to the reader? Many tables should either be placed in the supplementary files or converted to figure, such as bar plots showing the confidence interval or heatmap to filter on only significant results. The authors could also consider using more color in their figures.

The paper could benefit more with using in-depth statistical analysis in addition to simple frequency counts or percentage of occurrence. For example, they could highlight key results by putting more emphasis on the comparison of the regional, national, and global levels at species/family/order levels since this comparison allows for statistical analysis and put the numbers in perspective. The statistical analysis to be done could also link to a hypothesis to be tested. So far, this is missing in the paper, or at least, the framework where the analysis is anchored.

The tables and figures are consecutively shown which make it difficult to follow the text. If not necessary for the discussion of the results, better to put some in the supplementary files.

Specific comments:

Lines 20-28 (Abstract) – A lot of statistics are enumerated, which can be found in the Results. Instead, try to summarize similarities, highlight the differences and the significant findings, and why these findings are important. The abstract should entice the reader to continue reading.

Line30 - define what the abbreviation IUCN means

Lines 157-158 – Explain in detail the chi-square test (e.g. which one is the expected and the observed data, etc.). Is Chi-square test the right statistical method or the two sided T-test? Alternatively, check if you can do more fine-grained statistics. Would it be possible to do statistics to see if there are significant differences between species, family, or order of fish among the regional, national, and global levels? Can heatmap or side-by-side bar plot be used to highlight any differences among the regional, national, and global levels at species, family, and order levels? If any interesting differences come up with the analysis, link this to the discussion of conservation and management strategies to alleviate the problems on the regional level.

Pages 5-13 (Table 1): Too long to put in the main text. I suggest putting the Table 1 as supplementary material, since this is basically raw data and has been processed for succeeding figures. If placed in the main result, then the Table should be processed into Figure highlighting key findings.

Pages 14-15, Table 2 - Consider using side by side bar plots with confidence interval to show significance or to filter results on significance.

Page 16 (Figure 5) – Instead of showing the total relative abundance in conservation, use a stacked bar plot to show the conservation status per family of fish.

Page 18 (Table 4) – Consider using a heatmap or stacked bar plot to show what is relevant, order by level of being endangered or something. The figure can be linked to the table itself to supplementary material.

Discussion – Some parts of the discussions are a repetition of what are mentioned in the results (e.g. 302-304; 319-321; 350-352; 394-398). I miss in the discussion part that answer  the second of the aim of the study, which is the conservation strategies for the management of threatened species.

Line 414 – Make the heading “Conclusion” only.

Lines 423- 443 – In connection with comment in Line 414 and the discussion, the conservation and management strategies in lines 423-443 should be part of the discussion. These recommendations should be link to the finding in the results to make a nice story. E.g. Based on the results, which species/family/order of fishes are more vulnerable or critically endangered and what (current) fishery management strategies need to be adjusted or implemented to address the current conservation status.

As replacement for lines 423-443 summarize shortly the recommendations that are elaborated in the discussion. Then add the limitations of the study, methods etc. and the future direction of research.

Author Response

Dear Reviewer,

Reviewer 2 Report

Major comments

The introduction of the work brings a lot of information and different values to talk about fish biodiversity in the world and in Bangladesh. It is often unclear what these values are referring to. Furthermore, subjects often come and go, which makes reading difficult, for example when the author starts the introduction by talking about biodiversity, after the impacts and then goes back to talking about species richness and in the end impacts again. I suggest reviewing the introduction and organizing the sections to give a greater flow in the reading and avoid a "back and forth" in the subjects.

The methodology is a little confusing. I suggest rewriting it to explain in a better way how the collections were made. As I understand it, collections were carried out in markets and ports and experimental collections in addition to questionnaires. I suggest separating these three forms of collection and explaining how each one was carried out. If you performed collections only in ports, markets, and landing centres, your sample is biased because it cannot be said that you are evaluating the species richness of a certain area, but the species that have commercial value for fisheries. This must be explained very well. The methodology needs to be better described, it is still not clear how the collection was carried out. Since collections were made in different districts, why was the richness between districts not analyzed and discussed?

The discussion is poor and many times it is just a repetition of the results, comparing the results obtained in this work with results obtained by other works but without discussing the implications of these results for management and conservation.

The conclusions and recommendations are not linked to the results obtained. This paragraph is just a set of generic recommendations that can be used in different situations, but in this case, it is not linked with the results and much less with the discussion.

I suggest a major change in these paragraphs and that the implication of the results for the management and conservation of fish fauna be really discussed.

I am not a native English speaker, but this English is quite unique and needs major improvements.

Minor comments

Line 50: “2513” maybe would be 2,513”

Line 51: “11952” maybe would be 11,956”. Note the use of commas and periods throughout the manuscript and correct them.

Line 51 - 52: There are three numbers that refer to the number of freshwater fish species. It is not understood what they mean and why they are three different values. Rearrange and explain better.

Line 57: “hoars, baors, beels” are they local words? If so, describe them and put them in italic.

Line 58: Replace “man-made” with “artificial”

Line 58: Does the number 253 refer to all fish species found in Bangladesh? If it is a list of species evaluated by the IUCN this does not necessarily mean that the number of species is restricted to that.

Line 58 – 60: Perhaps the paragraph will be better if you put the total number of species and then discriminate which ones are freshwater, brackish and/or marine if that is the case.

Line 67 - 70: Insert the acronyms for the words, Critically Endangered, Endangered and Vulnerable, Extinct or Regionally Extinct category, as you did for the other risk categories in lines 68 and 69.

Line 74: What do you mean by “mother fishery”? breeding or nursery fishing grounds?

Line 75: What is “MT”?

Line 76: “0.152 million registered fishermen”, replace with the whole number. Are they all men? If not replace "fishermen" with "fishers"

Line 104: In the text is arat/wholesale fish market and in the map is Aroths/wholesale. Correct this.

Line 105 – 106: How were these sampling points defined? what is the sampling period? How were these samples taken? What is the fishing gear used to carry out these samplings? were carried out in rivers, streams, lakes? It is not enough to just talk about the number of samples and the districts, you must talk about the whole process of how the samples were carried out.

Lines 116 – 121: It is still unclear to me how this data collection was carried out. Are there experimental collections or was it just a survey of which species occur in the region? How was this collection or surveys carried out? Isn't there the possibility of misidentification using only questionnaires? The inclusion of questionnaires used as supplementary material could help.

Line 138: what does “viz.” means?

Line 139 – 142: You said that the commercial value was defined according to IUCN Bangladesh in lines 130 – 131.

Line 145: viz.

Line 164: why did you call "Table 1", on line 164, and then "Table 3", on line 171? Where is "Table 2"?

Line 176: viz

Line 182: What does SIS mean? You cannot insert acronyms in the text without explaining them first.

Line 183: You call “figure 4” to talk about SIS (?) composition, but figure 4 is about size, trophic and habitat. 

Line 183: Why did you first call "Figure 4", on line 183, and then call "Figure 3", on line 185?

Line 196: Table 2 was not called in the main manuscript text.

Line 209: Merge topics 3.2 and 3.3.

Lines 212 – 214: Avoid repeating information that is already on the chart.

Lines 223 – 227: why did you use percentages and whole numbers in section 3.3 and not in section 3.2? Standardize and avoid repetitions of information already included in the charts.

Lines 300 – 301: Rearrange the information in this paragraph.

328: In line 164 you say 176 sp. were collected. And here you say 175.

333: Here you say 5 in number, and on line 335 seven and nine are spelt out in full. Choose a shape and standardize it for the entire manuscript.

378: Replace “In” with “in”

380: “According to NAME OF THE AUTHOR [10] 

378: You just call “CR” to refer to a risk category but in line 388 you spelt out in full. Choose one way and standardize it for the entire manuscript. 

388: You spelt the full name of the specie and in line 391 used an abbreviation. Choose one way and standardize it for the entire manuscript.

398: “IUCN (2015)” correct it.

Author Response

Dear Reviewer, 

Round 2

Reviewer 1 Report

The paper has significantly improved after revision. Below are some minor revisions as condition for approval of acceptance of this paper.  

1. Please make sure to remove redundancy in text as seen in the following line numbers (e.g. lines 293-295; 377-381; 417-419; 423-427; 480-481, etc.). Check throughout the document since there are still more.

2. Under methodology, 2.2., the authors mentioned ‘People’s perception’. Do you mean respondent’s perceptions? If so please consider revising.

3. Instead of “named”, use “namely”.

4. Lines 162-170 is repeated again in lines 171-178. Correct this.

5. Line 502. Connect the sentence to line 503.

6. Conclusion: After the first sentence, summarize the important findings based on the discussion part. For example,  Fish species, such as Cypriniformes and siluriformes are the ones facing significant threat level; Some species that are critically endangered at national and global levels are intensively being cultured, etc. Through this, the authors can make connection to the summarized recommendations.

7. Make sure that the abstract in the revised text is used and not the one on the online submission form (old abstract).

Reviewer 2 Report

The author did the most changes suggested.

Author Response

During the second review, the reviewer did not suggest any alterations.